# Regulation and Function of C-Type Natriuretic Peptide (CNP) in Gonadotrope-Derived Cell Lines

**DOI:** 10.3390/cells8091086

**Published:** 2019-09-14

**Authors:** Samantha M Mirczuk, Andrew J Lessey, Alice R Catterick, Rebecca M Perrett, Christopher J Scudder, Jordan E Read, Victoria J Lipscomb, Stijn J Niessen, Andrew J Childs, Craig A McArdle, Imelda M McGonnell, Robert C Fowkes

**Affiliations:** 1Endocrine Signalling Group, Royal Veterinary College, University of London, Royal College Street, London NW1 0TU, UK; samantha.byers@admin.cam.ac.uk (S.M.M.); andrewlessey@xstrahl.com (A.J.L.); alicecatterick@hotmail.com (A.R.C.); cscudder@rvc.ac.uk (C.J.S.); jordaneread89@hotmail.co.uk (J.E.R.); 2Laboratories for Integrative Neuroscience and Endocrinology, Department of Clinical Sciences at South Bristol, University of Bristol, Whitson Street, Bristol BS13NY, UK; bexperrett@gmail.com (R.M.P.); craig.mcardle@bristol.ac.uk (C.A.M.); 3Comparative Biomedical Sciences, Royal Veterinary College, University of London, Royal College Street, London NW1 0TU, UK; a.childs@imperial.ac.uk (A.J.C.); imcgonnell@rvc.ac.uk (I.M.M.); 4Clinical Science and Services, Royal Veterinary College, Hertfordshire AL9 7TA, UK; vlipscomb@rvc.ac.uk (V.J.L.); sniessen@rvc.ac.uk (S.J.N.)

**Keywords:** natriuretic peptides, gonadotropin releasing hormone (GnRH) signaling, gonadotrope, gene expression, cGMP

## Abstract

C-type natriuretic peptide (CNP) is the most conserved member of the mammalian natriuretic peptide family, and is implicated in the endocrine regulation of growth, metabolism and reproduction. CNP is expressed throughout the body, but is particularly abundant in the central nervous system and anterior pituitary gland. Pituitary gonadotropes are regulated by pulsatile release of gonadotropin releasing hormone (GnRH) from the hypothalamus, to control reproductive function. GnRH and CNP reciprocally regulate their respective signalling pathways in αT3-1 gonadotrope cells, but effects of pulsatile GnRH stimulation on CNP expression has not been explored. Here, we examine the sensitivity of the natriuretic peptide system in LβT2 and αT3-1 gonadotrope cell lines to continuous and pulsatile GnRH stimulation, and investigate putative CNP target genes in gonadotropes. Multiplex RT-qPCR assays confirmed that primary mouse pituitary tissue express *Nppc,*
*Npr2* (encoding CNP and guanylyl cyclase B (GC-B), respectively) and *Furin* (a CNP processing enzyme), but failed to express transcripts for *Nppa* or *Nppb* (encoding ANP and BNP, respectively). Pulsatile, but not continuous, GnRH stimulation of LβT2 cells caused significant increases in *Nppc* and *Npr2* expression within 4 h, but failed to alter natriuretic peptide gene expression in αT3-1 cells. CNP enhanced expression of *cJun*, *Egr1*, *Nr5a1* and *Nr0b1*, within 8 h in LβT2 cells, but inhibited *Nr5a1* expression in αT3-1 cells. Collectively, these data show the gonadotrope natriuretic peptide system is sensitive to pulsatile GnRH signalling, and gonadotrope transcription factors are putative CNP-target genes. Such findings represent additional mechanisms by which CNP may regulate reproductive function.

## 1. Introduction

The natriuretic peptides are a highly conserved group of peptide hormones, comprising of Atrial-, B-type and C-type natriuretic peptides (ANP, BNP and CNP, respectively). They are structurally related, sharing a common 17-membered disulphide ring, which imparts biological activity at their respective particulate guanylyl cyclase receptors, GC-A and GC-B [1,2,3]. Homology between species is high, with CNP being the most conserved member of the family, where there is greater than 90% similarity between piscine and human sequences [4].

Both ANP and BNP are expressed in many peripheral tissues, yet their highest concentrations are found in cardiac atria and ventricles, and have profound effects on cardiac output and cardiovascular function [3]. In contrast, the tissue distribution of CNP is, perhaps, the broadest of the three major mammalian natriuretic peptides, and ranges across virtually all endocrine tissues, bone, the CNS and endothelial cells [1,2,3]. Initial studies that detailed the expression profile of CNP found relatively high concentrations within the anterior pituitary gland [5,6], and pituitary gonadotropes were subsequently shown to be the predominant endocrine cell lineage to express CNP [7]. Our subsequent pharmacological and molecular investigations revealed gonadotropes to be major sources of both expression and function for CNP in the pituitary [8], and also demonstrated expression of both CNP and GC-B in normal human fetal pituitaries and a range of pituitary adenomas [9]. The elegant mouse models of CNP/GC-B disruption not only revealed severe achondroplasia and early death, but also suggested impaired fertility and reduced growth hormone secretion, phenotypes that strongly implicate a pituitary role for CNP/GC-B signalling [10,11].

Gonadotropes are regulated by numerous endocrine and paracrine factors, the principal ones being the gonadotrophin releasing hormone (GnRH) as well as the gonadal steroids [12,13,14]. Previous studies have shown that the expression of natriuretic peptides and their receptors are sensitive to changes in gonadal steroids and gonadotropins. In the uterus, estradiol rapidly induced CNP expression [15], whereas equine chorionic gonadotropin (eCG)-treated mouse ovaries showed elevated ANP, CNP, GC-A and GC-B expression [16]. Our recent studies have shown that the proximal murine *Nppc* promoter is stimulated by chronic GnRH treatment, in a calcium and protein kinase C-dependent manner [8], and transcription of both the *Nppc* and *NPR2* genes appears to involve the Sp1/Sp3 family transcription factors [8,9]. At the functional level, GnRH and CNP appear to reciprocally antagonize their respective signaling pathways, as GnRH causes heterologous desensitization of GC-B receptors and cGMP signaling [17,18] whereas CNP attenuates GnRH-stimulated calcium mobilization in gonadotrope cell lines [19]. Despite these observations, CNP fails to significantly alter the secretion of LH from primary rat pituitary cells, but does stimulate the transcriptional activity of the human glycoprotein hormone α-subunit gene promoter in LβT2 cells [8,17]. Thus, the role of CNP in gonadotrope function still remains somewhat enigmatic.

The vast majority of historical investigations of GnRH signalling in vitro have ignored the physiological manner in which GnRH is usually secreted from the hypothalamus; in pulses. After the initial observation which characterised the role of pulsatile GnRH in male rats [20], more recent studies have highlighted the importance of utilizing a more physiologically relevant GnRH treatment paradigm, which has been illustrated by several studies reporting differential effects of continuous versus pulsatile GnRH on both gonadotrope gene expression and in terms of signalling responses to GnRH [21,22,23,24,25]. Our own studies that investigate signalling events downstream of the GnRH receptor, have clearly established relationships between GnRH pulse frequency and transcriptional output [26,27,28,29]. However, despite knowing that gonadotropes are likely target cells for CNP, that GnRH and CNP are reciprocally antagonistic in their signaling in gonadotrope cell lines, and that GnRH can activate the *Nppc* promoter [8,18,19], the potential relationship between pulsatile GnRH and natriuretic peptide expression in gonadotropes has not been investigated.

The biological effects of natriuretic peptides are, overwhelmingly, mediated by their capacity to increase the levels of cGMP in their target tissues [1,3,4]. Although the regulation of gene expression by cGMP has been reported in many systems [30,31,32,33,34], putative target genes for natriuretic peptide action in the pituitary have yet to be identified. Here, we investigate the sensitivity of the gonadotrope natriuretic peptide system to pulsatile GnRH stimulation, and identify novel transcriptional targets for CNP.

## 2. Materials and Methods

### 2.1. Materials

GnRH, CNP-22 (referred to as CNP) and all other chemicals were purchased from Sigma (Sigma-Aldrich, Poole, UK) unless otherwise stated.

### 2.2. Cell Culture

LβT2 and αT3-1 gonadotrope cells were grown in monolayer culture in DMEM supplemented with high glucose (4500 mg/L) containing 10% (*v/v*) FCS, 1% (*v/v*) antimycotic/antimicrobial, as previously described [8]. Cells were passaged twice weekly and incubated at 37 °C in a humidified 5% (*v/v*) CO_2_/95% (*v/v*) air incubator. For experiments, cells were plated at 1 × 10^6^ cells/well in 6-well plates and allowed to adhere, prior to serum starvation in DMEM supplemented with 1% (*w/v*) BSA overnight. In some experiments, cells were treated in a pulsatile manner for four hours, whereby cells were exposed to 0 or 100 nM GnRH for five minutes, every hour, after which the treatments were removed and the cells washed three times with PBS, before being returned to the incubator in DMEM containing 1% (*w/v*) BSA. In addition, some cells were continuously exposed to 0 or 100 nM GnRH for four hours. At the end of the experiments, spent media were removed and cells were washed, prior to extraction of total RNA using RNABee (AMS Biotechnology, Abingdon, UK). For CNP experiments, cells were treated with 0 or 100 nM CNP for up to 24 h prior to RNA extraction. All concentrations for treatments were selected on the basis of previously published maximal effects [8,35].

### 2.3. Tissue Collection, RNA Extraction and Multiplex GeXP RT-qPCR Assay

Tissue was collected from 5 to 8 C57/B6 male and female mice in accordance with UK Home Office Guidelines (PPL70/6965), placed into either a 1.5 mL Eppendorf tube or foil (brain and heart tissue) and snap frozen in liquid nitrogen. Total RNA was extracted from primary mouse tissue (cardiac ventricle, adrenal, adipose, brain, pituitary, kidney, liver and testis), or from cultured αT3-1 or LβT2 cells, using RNAbee reagent, and subjected to DNase treatment (Qiagen, Poole, UK), as described previously [8]. RNA concentrations were determined using ND-100 spectrophotometer (Nanodrop, Thermo Fisher, Hemel Hempstead, UK). Two customised GeXP multiplex assays were designed, to detect natriuretic peptide gene targets (*Nppa*, *Nppb*, *Nppc*, *Npr1*, *Npr2*, *Npr3*, *Corin* and *Furin*), or gonadotrope transcription factor genes (*Nr5A1*, *Nr0B1*, *cJun*, *cFos* and *Egr1*) (Appendix A). In all assays, 100 ng of total RNA was used per sample. Target-specific reverse transcription and PCR amplification was performed as previously described [36] and in accordance with manufacturer’s instructions (Beckman Coulter, High Wycombe, UK). In brief, a master mix was prepared for reverse transcription reactions as detailed in the GeXP Starter Kit (AB Sciex, Warrington, Cheshire, UK), and performed using a G-Storm GS1 thermal cycler, using the programme protocol: 48 °C for 1 min, 42 °C for 60 min, and 95 °C for 5 min. From this, an aliquot of each reverse transcription reaction was added to PCR master mix containing GenomeLab kit PCR master mix (AB Sciex, Warrington, Cheshire, UK), and Thermoscientific Thermo-Start Taq DNA polymerase (Thermo Fisher; AB Sciex, Warrington, Cheshire, UK). PCR reaction was performed using a 95 °C activation step for 10 mins, followed by 35 cycles of 94 °C for 30 s, 55 °C for 30 secs and 70 °C for 60 secs. Products were separated and quantified using the GeXP CEQ^TM^ 8000 Genetic Analysis System AB Sciex, Warrington, Cheshire, UK), and GenomeLab Fragment Analysis software (eXpress Analysis Version 1.0.25, Beckman Coulter, Inc.).

### 2.4. Data Presentation & Statistical Analysis

Data shown are means ± SEM from individual RNA extractions, pooled from experiments performed in duplicate or triplicate. αT3-1 and LβT2 cells were used from a range of passages, between 10 and 35. Numerical data were subjected to ANOVA, followed by Tukey’s or Dunnett’s multiple comparison tests (where appropriate), accepting *p* < 0.05, using in-built equations in GraphPad Prism 7.0a for Mac (GraphPad, San Diego, CA, USA).

## 3. Results

### 3.1. Expression Profiling of the Natriuretic Peptide System in Primary Mouse Endocrine Tissues by Multiplex RT-qPCR

Our previous studies have identified an intact, and functional, natriuretic peptide system in gonadotrope cell lines, mouse and rat pituitaries, and a range of human pituitary adenomas [7,8,9,17,18]. However, these qualitative studies in pituitary cells lines and pituitary tissue did not examine all components of the natriuretic peptide system (such as the associated convertase enzyme genes), nor did it compare expression of these components in other endocrine tissues. Therefore, we utilised multiplex RT-qPCR assays to examine the expression of natriuretic peptide-associated genes. Total RNA was extracted from heart, adrenal, adipose, brain, pituitary, kidney, liver, testis and ovarian tissue collected from C57/B6 mice. Subsequently, target-specific reverse transcription was performed, followed by qPCR and quantitative analyses using the GeXP Genetic Analysis System, as we have described previously [36]. To confirm that all primer sets within the assay were functional, we performed an initial screen of murine cardiac tissue samples. As shown (Figure 1A), the electropherogram confirmed expression of each gene of interest included within the assay. Negative control samples were run routinely and failed to detect any transcripts, confirming the specificity of the multiplex RT-qPCR assay.

We next examined the expression of natriuretic peptide associated genes in a range of endocrine tissues from both male and female C57/B6 mice. As shown (Figure 1B; Appendix A), differential expression of transcripts for natriuretic peptides (*Nppa, Nppb* and *Nppc*), receptors (*Npr1, Npr2* and *Npr3*) and convertase enzymes (*Corin* and *Furin*) was seen across endocrine tissues, with only heart, brain and testis expressing detectable transcripts for all gene targets within the multiplex assay.

### 3.2. Expression Profiling of the Natriuretic Peptide System in αT3-1 and LβT2 Gonadotrope Cell Lines

Having used the primary mouse tissues to optimise the multiplex RT-qPCR assays, total RNA was then generated from αT3-1 and LβT2 cells, prior to analyses using the multiplex RT-qPCR assays. As expected, specific transcripts were detected for *Nppc*, *Npr1*, *Npr2*, *Npr3* and *Furin* in both cell lines (Figure 2), with gene expression levels of *Npr2* being significantly less in LβT2 compared with αT3-1 cell line (* *p* = 0.02). *Corin* was detected in αT3-1 but not LβT2 cells. In keeping with primary mouse pituitary tissue, transcripts for *Nppa* and *Nppb* were absent from both cell lines.

### 3.3. Effect of Continuous or Pulsatile Exposure to GnRH on Natriuretic Peptide Gene Expression

We next investigated whether the expression of the natriuretic peptide genes was sensitive to GnRH stimulation. To do this, we employed a more physiologically relevant treatment paradigm, by comparing the effects of either continuous (4 h) or pulsatile (5 min/h for 4 h) GnRH administration (Figure 3A). In LβT2 cells (Figure 3B), *Nppc*, *Npr2* and *Npr3* were significantly up-regulated by pulsatile GnRH treatment (by 2.2 ± 0.2-fold, 1.7 ± 0.2-fold and 1.8 ± 0.2-fold, for *Nppc*, *Npr2* and *Npr3* respectively; ** *p* < 0.01). In contrast, only *Npr3* was upregulated by continuous GnRH administration (by 2.2 ± 0.2-fold, **** *p* < 0.0001). GnRH failed to alter gene expression of any of these transcripts in αT3-1 cells, regardless of continuous or pulsatile delivery (Figure 3C).

To ensure that the lack of responsiveness to GnRH of natriuretic peptide gene transcripts in αT3-1 cells did not indicate an artifactual observation, we performed additional multiplex RT-qPCR assays to quantify changes in relevant gonadotrope transcription factors (*cFos*, *cJun*, *Egr1*, *Nr5a1*, *Nr0b1*). As shown (Figure 4A), transcripts for all five transcription factors were detected in both cell lines, albeit at differing expression levels. Analyses of the same GnRH-treated LβT2 samples revealed differential responsiveness of *cFos* and *Egr1* transcripts to continuous and pulsatile GnRH (Figure 4B); *cFos* was increased by 3.6 ± 0.2-fold (**** *p* < 0.0001) and 2.5 ± 0.5-fold (** *p* < 0.01; continuous and pulsatile GnRH, respectively); *Egr1* expression was increased by 2.7 ± 0.2-fold (**** *p* < 0.0001) and 1.6 ± 0.2-fold (**** *p* < 0.0001; continuous and pulsatile GnRH, respectively). Gene expression changes were similar when the same αT3-1 samples were analysed, with both *cFos* and *Egr1* transcripts showing differential responsive need to GnRH treatment (*cFos* increased by 2.0 ± 0.2-fold (* *p* < 0.05) and 2.2 ± 0.2-fold (** *p* < 0.01) with continuous and pulsatile GnRH treatment, respectively; *Egr1* increased by 1.9 ± 0.1-fold (**** *p* < 0.0001) with continuous GnRH exposure. Collectively, these data suggest that components of the natriuretic peptide system are differentially sensitive to GnRH treatment in LβT2 cells, dependent upon the pattern of stimulation.

### 3.4. CNP Effects on Expression Levels of Gonadotrope Transcriptional Regulators and Signaling Genes

Although the pituitary was identified as a major site of production for CNP [5], the biological function of this peptide in anterior pituitary cells is still poorly understood. As *Nppc* was the only natriuretic peptide transcript detected in both gonadotrope cells lines and in primary mouse pituitaries, we focused our examination of natriuretic peptide affects to gonadotrope gene expression on CNP alone. LβT2 and αT3-1 cells were treated with 100 nM CNP (maximally effective at stimulating cGMP accumulation, [8,18]) for up to 24 h, before extracting total RNA and analyses using the gonadotrope transcription factor multiplex assay. As shown (Figure 5), in LβT2 cells, CNP significantly stimulated *cJun* (by 1.5 ± 0.1-fold, *** *p* < 0.001, 8 h), *Egr1* (by 2.1 ± 0.1-fold, **** *p* < 0.0001, 24 h), *Nr5a1* (by 1.8 ± 0.1-fold, *** *p* < 0.001, 24 h), and *Nr0b1* (by 1.4 ± 0.1-fold, ** *p* < 0.01, 8 h) expression. In marked contrast, CNP failed to alter the expression of *cFos, cJun, Egr1* or *Nr0b1* expression in αT3-1 cells, and caused a significant reduction to *Nr5a1* (by 0.25 ± 0.02-fold, **** *p* < 0.0001, 24 h) transcript expression. Together, these data identify the first transcriptional targets for CNP signalling in the more mature, LβT2 gonadotrope lineage cells.

## 4. Discussion

Initial expression profiling studies identified the anterior pituitary as a major source of CNP [5], with subsequent investigations highlighting the importance of pituitary gonadotrope cells [8,17,18]. Despite limited data describing a relationship between CNP and GnRH signalling [8,17,18], very little is understood with regards to the regulation and role of CNP in pituitary gonadotropes, although there is much greater understanding of the role of CNP in other parts of the hypothalamo–pituitary–gonadal (HPG) axis [4,37,38,39,40]. In the current study, we have used multiplex RT-qPCR technology to conduct a comprehensive expression profile of natriuretic peptide-associated genes in primary mouse endocrine tissues, as well as the widely used αT3-1 and LβT2 gonadotrope-derived cells. In addition, we have identified the first potential transcriptional targets of CNP action in gonadotrope-derived cells.

The use of multiplex RT-qPCR technology allows the detection of multiple gene transcripts from a single sample [36,41,42]. Our expression profiling of murine endocrine tissues revealed that all components of the natriuretic peptide system were ubiquitously expressed, apart from *Nppa* and *Nppb* (which were predominantly expressed in the heart). These findings are in keeping with CNP performing an autocrine/paracrine role throughout the periphery, as well as the CNS [4,43]. The failure to detect either *Nppa* or *Nppb* transcripts in the pituitary was mimicked when expression profiling was performed in the two gonadotrope-derived cell lines, again supporting previous studies that indicate CNP to be the predominant pituitary natriuretic peptide [7,8,9,18]. The relative abundance of *Furin* in both primary pituitary tissue and in LβT2 and αT3-1 cells, also suggests that gonadotrope-lineage cells are primed to process CNP, as opposed to ANP/BNP (as *Corin* was either absent, or expressed at very low levels). The only transcript which showed differential expression in the gonadotrope cell lines was *Npr2*, which was significantly less abundant in LβT2 cells. However, this supports our previous pharmacological profiling of particulate guanylyl cyclase activity in these cells [8].

Endocrine control of natriuretic peptide gene expression has been shown in several tissues, and involves transcriptional regulation by gonadal steroids and peptide hormones [44,45,46,47]. Our previous studies suggested that the *Nppc* promoter could be activated by GnRH in αT3-1 cells [8], via a combination of calcium and MAPK signalling pathways. The data described herein are the first descriptions to show regulation of endogenous *Nppc* expression in response to GnRH signalling, whereby pulsatile GnRH significantly increased *Nppc*, *Npr2* and *Npr3* expression in LβT2 cells, without altering expression in αT3-1 cells. A possible explanation for these differences in responsiveness could be a reflection of the stimulation paradigm. Firstly, early studies using gonadotrope cell lines to investigate the effects of GnRH pulsatility used a priming protocol [48], whereas our paradigm has used pulse-naive LβT2 and αT3-1 cells. Secondly, in the current study we have examined gene expression changes at the end of a 4hr protocol, to better reflect primary transcriptional effects of GnRH; but this would fail to take account of any longer term/secondary transcriptional effects. Furthermore, the pattern of the pulsatile GnRH administration (5 min/h) is slightly longer than would be expected in vivo, which might result in altered decoding of the GnRH effect. Nevertheless, the robust *Nppc* response to pulsatile GnRH is further evidence that GnRH is a regulator of natriuretic peptides in gonadotrope derived cells, supporting previous studies that have implicated just such a relationship [7,8,17]. As the major role of *Npr3*/NPR-C is to clear natriuretic peptide from the circulation [1] and reduce bioavailability, it is surprising that both continuous and pulsatile GnRH administration caused a significant increase in both *Npr3* transcripts as well as *Nppc* and *Npr2*. However, this may simply represent a normal physiological upregulation in *Npr3* to counteract the increase to the CNP/GC-B pathway.

The lack of any natriuretic peptide gene expression response to GnRH in αT3-1 cells might have reflected a loss of overall responsiveness in this cell line. Expression of *Gnrhr* was not affected by continuous or pulsatile exposure to GnRH in either cell line (Appendix A). Furthermore, when examining the transcriptional response of key gonadotrope transcription factors, both *cFos* and *Egr1* were significantly up-regulated in both cell lines, as shown previously [21,29,49], confirming that an appropriate signalling response to GnRH in these αT3-1 cells was intact. Downstream activation of MAPK signalling pathways following GnRH signalling has been identified as a major mechanism of action in gonadotrope cells [14,50,51,52,53,54], often providing the membrane-to-nucleus link to control gene transcription. In the current study, we have not identified the likely mechanism by which GnRH affects *Nppc* and *Npr2* expression. However, given the sensitivity of the *Nppc* promoter to calcium/MAPK stimulation [8], and the wealth of literature linking pulsatile GnRH signalling to MAPK activation [16,55], it is possible that these pathways are at least partially involved in the control of endogenous *Nppc* and *Npr2* expression in gonadotrope cells.

Despite being expressed at relatively high tissue concentrations in the anterior pituitary, the role of CNP in pituitary function has remained unclear [4]. CNP is the most potent activator of cGMP production in the anterior pituitary and gonadotrope cell lines, whilst failing to alter gonadotropin secretion from rat pituitary cells [8,17,18]. Our identification of putative CNP-target genes in LβT2 and αT3-1 cells represents the first biological role for CNP beyond simply elevating cGMP production. Interestingly, the gene expression response to CNP was different in the two cell lines; CNP stimulated expression of *cJun*, *Egr1*, *Nr5a1* and *Nr0b1* in LβT2 cells, but reduced *Nr5a1* expression in αT3-1 cells. These cell lines represent gonadotrope cells at different stages of pituitary development; αT3-1 cells being more representative of a gonadotrope progenitor cell, and LβT2 cells closely resembling the more mature gonadotrope [56,57]. It would be tempting to speculate that the differential gene expression response to CNP is associated with a differential developmental role for CNP in gonadotropes. Whilst there is some evidence in the CNS of developmental changes to the natriuretic peptide system [58], we have not specifically addressed whether this occurs in the pituitary. However, we have previously described the presence of an intact natriuretic peptide system (*NPPC*, *NPR2*) in human fetal pituitaries, normal adult pituitaries, and in human pituitary adenomas, regardless of origin [9]; as these tissues represent a broad range of developmental ages (fetal to adult), the lack of any substantial difference in the expression of *NPPC* or *NPR2* would suggest that developmental changes to CNP/GC-B in the anterior pituitary do not occur.

The mechanisms by which cGMP mediate gene expression are not fully elucidated, although both nitric oxide and natriuretic peptide signalling has been shown to have direct effects on the transcription of many genes [30,31,32,33,34]. For example, cGMP/PKG can drive *cFos* expression, via a number of response elements, including AP-1, CRE and an SRE [59]. cGMP signalling can also directly regulate CREB by activation of PKA, or indirectly, through the activation of MAPK [60]. As *cJun*, *Egr1*, *Nr5a1* and *Nr0b1* have all been shown to be transcriptionally regulated via the cAMP/PKA/CRE pathway [61,62,63,64], this represents a likely mechanism to explore in LβT2 cells. Additionally, we, and others, have previously shown that CNP can directly activate MAPK signalling in both rat pituitary tumour GH3 cells and LβT2 cells [65] or in melanoma cells [66]. Therefore, the differential effects of CNP on gene expression in LβT2 and αT3-1 cells could reflect differences in which signalling pathways are activated downstream of the GC-B receptor.

The role of CNP as an important regulator of gonadal and reproductive function is becoming increasingly clear, yet our understanding of how CNP influences other levels of the HPG remains enigmatic. Our data provide an indication of how CNP signalling might alter gonadotrope function, by controlling the expression of key transcription factors. As mutations to either *Nppc* or *Npr2* are linked with poor reproductive development and function, a thorough understanding of how CNP and cGMP regulate pituitary gonadotrope lines cells is important for the development of future fertility therapies.

## Figures and Tables

**Figure 1 cells-08-01086-f001:**
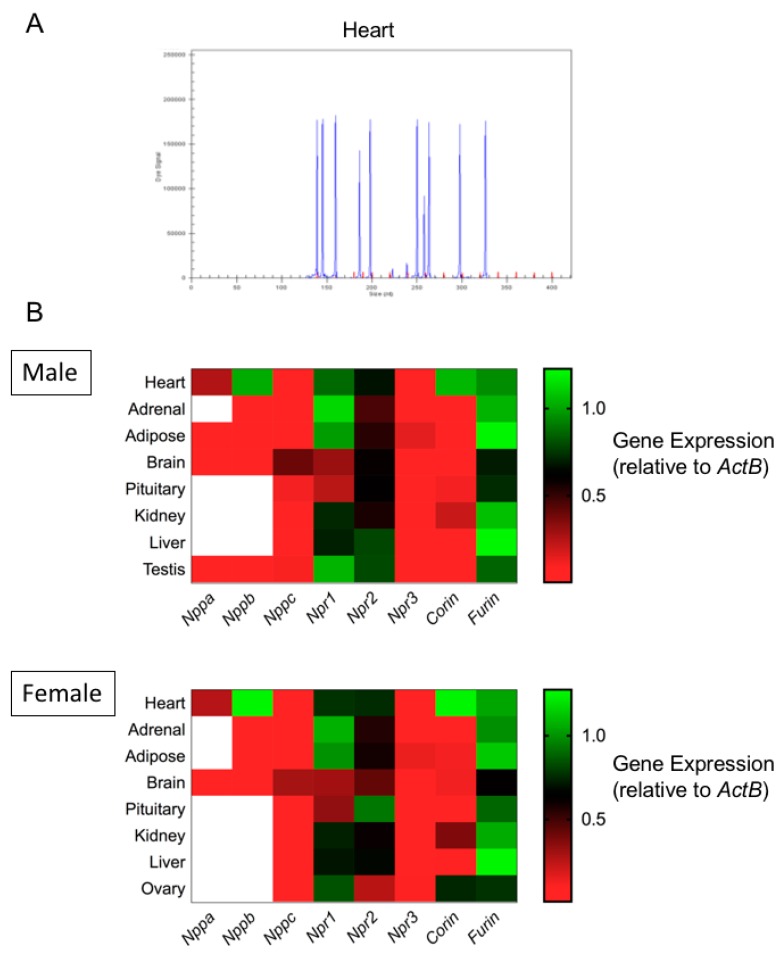
Expression profiling of the natriuretic peptide system in primary mouse endocrine tissues. (**A**) Murine mRNA sequences obtained from NCBI Nucleotide (https://www.ncbi.nlm.nih.gov/nuccore), were imported into express Designer Software (Beckman Coulter), from which multiplex primers were designed using the following parameters: maximum PCR product = 300 nt, minimum PCR product = 100 nt, minimum separation size = 7 nt. Multiplex PCR reactions were performed, using specific primers for *Nppa*, *Nppb*, *Nppc*, *Npr1*, *Npr2*, *Npr3*, *Furin*, *Corin*, *ActB*, *Gapdh* and *Rpl19* and an internal positive control KanR. As shown (Figure 1A), capillary gel electrophoresis was used to separate specific PCR products (blue peaks), and compared alongside the appropriate size standard (red peaks, 140–420 nt). (**B**) RNA was isolated from range of tissues from 12 week old male and female C57/B6 mice (heart, adrenal, adipose, brain, pituitary, kidney, liver, testis and ovaries; n = 5 to 8). Data shown are means (n = 5 to 8) of relative gene expression (normalized to *ActB*; red indicates low level expression, green indicates high level expression, white indicates no transcript detected).

**Figure 2 cells-08-01086-f002:**
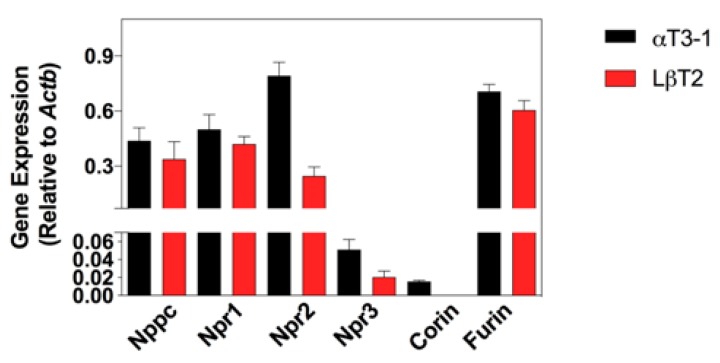
Natriuretic peptide expression profile in gonadotrope-derived cell lines. mRNA expression of natriuretic peptide components in untreated αT3-1 and LβT2 cells (*Nppa* and *Nppb* were not detected). Data shown are means ± SEM (n = 5 individual RNA extractions) of relative gene expression (normalized to *ActB*).

**Figure 3 cells-08-01086-f003:**
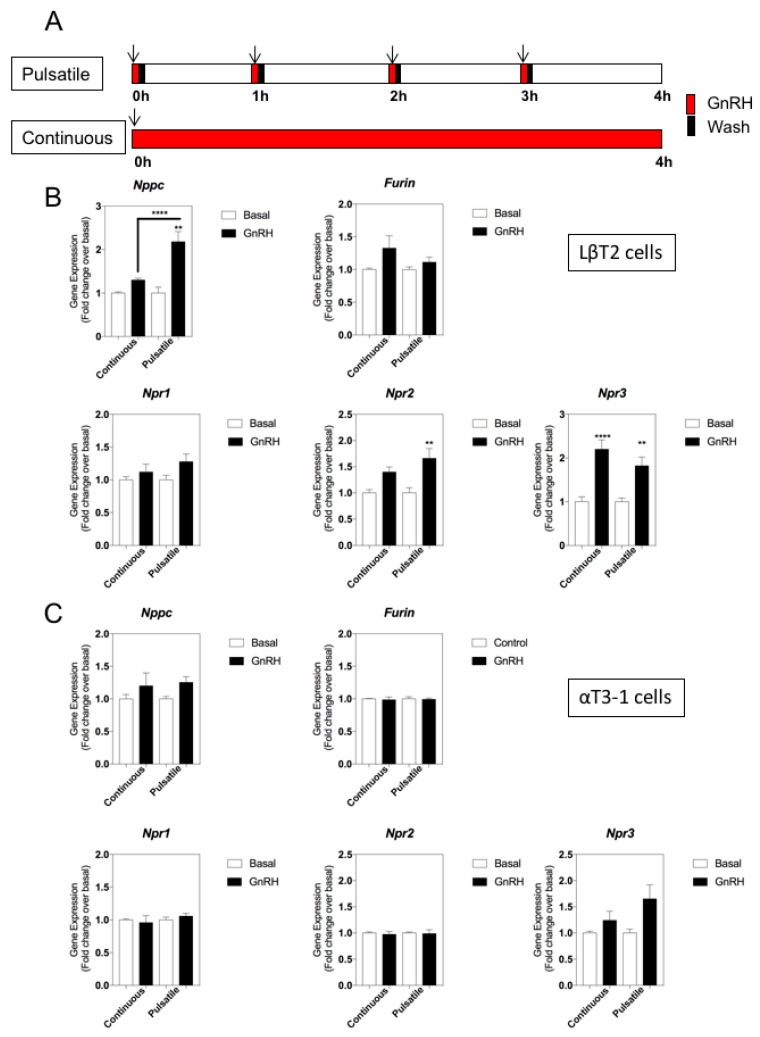
Effect of continuous or pulsatile GnRH stimulation on natriuretic peptide gene expression in LβT2 and αT3-1 cell lines. (**A**) Schematic of pulsatile or continuous GnRH treatment protocol, including wash periods. LβT2 cells (**B**) or αT3-1 cells (**C**) were treated with 0 or 100 nM GnRH, for either 4 h continuously, or as 5 min pulses every hour for 4 h, before extracting RNA and performing multiplex RT-qPCR to examine alterations in gene expression profiling of natriuretic peptide system (*Nppc*, *Furin*, *Corin*, *Npr1*, *Npr2*, *Npr3*). Data shown are means ± SEM (n = 6 to 9 individual RNA extractions) of relative gene expression (normalized to *ActB*); *** *p* < 0.001, ** *p* < 0.01, significantly different from basal, or continuous vs pulsatile, as indicated).

**Figure 4 cells-08-01086-f004:**
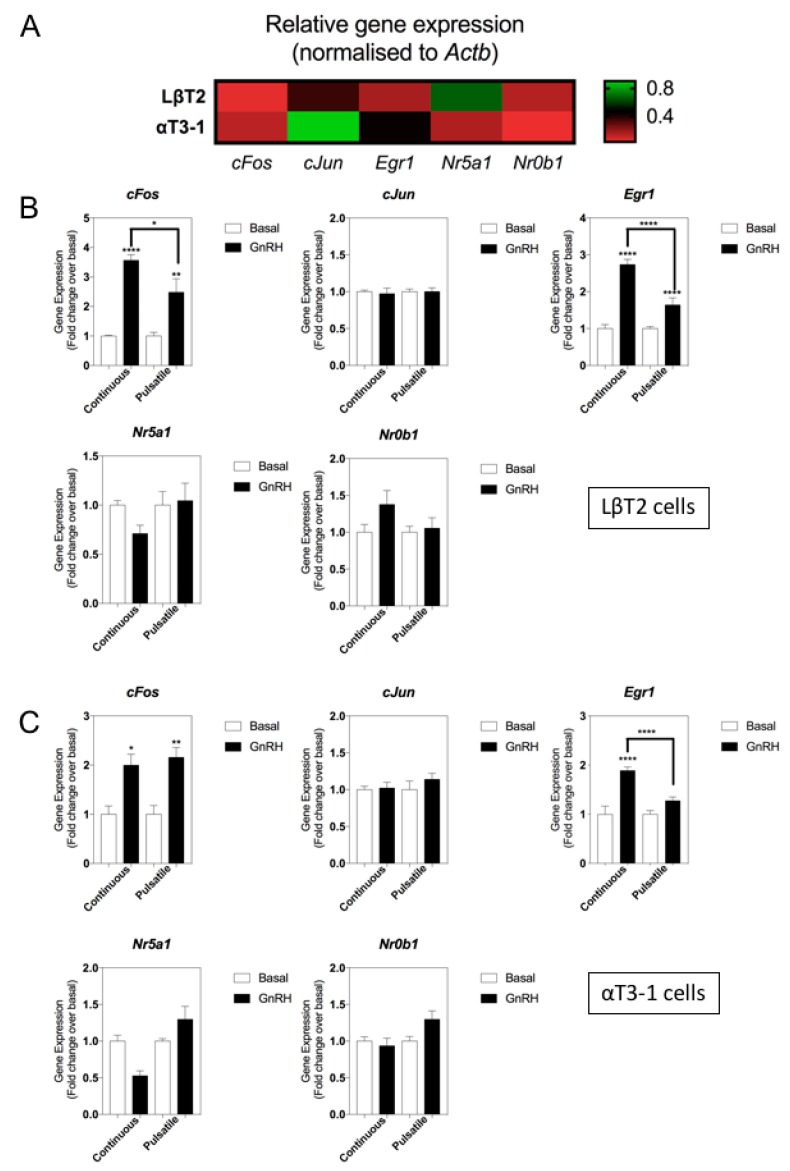
Effect of continuous or pulsatile GnRH stimulation on gonadotrope transcription factor gene expression in LβT2 and αT3-1 cell lines. (**A**) Comparison of gonadotrope transcription factor expression (*cFos*, *cJun*, *Egr1*, *Nr5a1*, *Nr0b1*) in LβT2 and αT3-1 cells. Data shown are means (n = 5 to 8 individual RNA extractions) of relative gene expression (normalized to *ActB*; red indicates low level expression, green indicates high level expression. (**B**,**C**) LβT2 cells (**B**) or αT3-1 cells (**C**) were treated with 0 or 100 nM GnRH, for either 4 h continuously, or as 5 min pulses every hour for 4 h, before extracting RNA and performing multiplex RT-qPCR for gonadotrope transcription factors. Data shown are means ± SEM (n = 6 to 9 individual RNA extractions) of relative gene expression (normalized to *ActB*); *** *p* < 0.001, ** *p* < 0.01, * *p* < 0.05, significantly different from basal, or continuous vs pulsatile, as indicated).

**Figure 5 cells-08-01086-f005:**
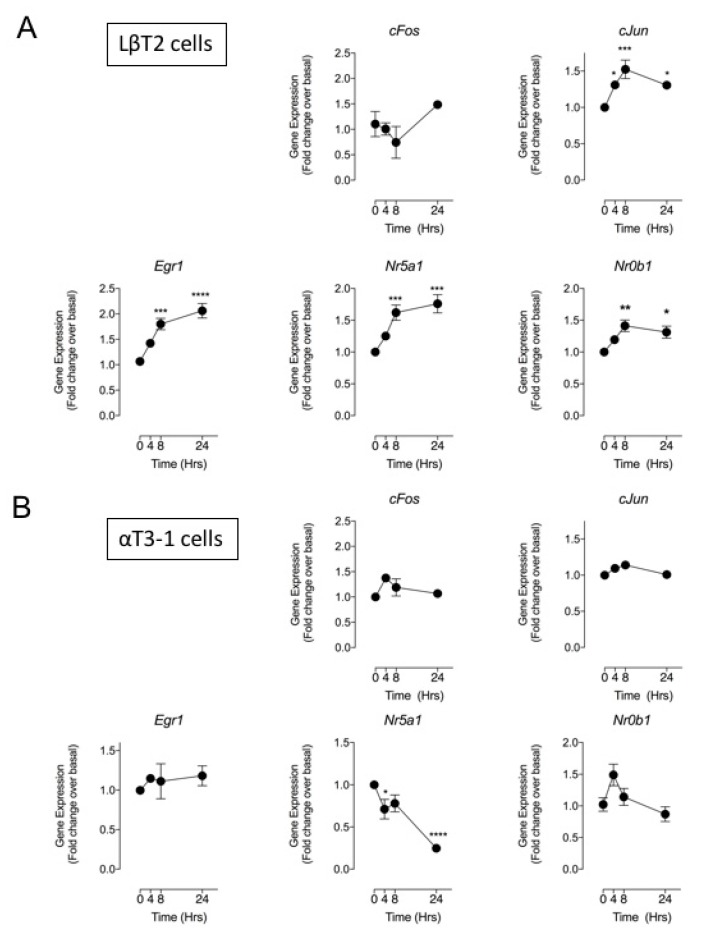
Effect of CNP on gonadotrope transcription factor gene expression in LβT2 and αT3-1 cells. LβT2 (**A**) and αT3-1 (**B**) cells were stimulated with 0 or 100 nM CNP for 0, 4, 8, and 24 h, before extracting RNA and performing multiplex RT-qPCR for gonadotrope transcription factors. Data shown are means ± SEM (n = 6 individual RNA extractions) of relative gene expression (normalized to *ActB*); **** *p* < 0.0001, *** *p* < 0.001, ** *p* < 0.01, * *p* < 0.05, significantly different from basal (0 h).

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
