# Peer review of "Regulation and Function of C-Type Natriuretic Peptide (CNP) in Gonadotrope-Derived Cell Lines"

_cells, 2019, doi:10.3390/cells8091086_

Round 1

Reviewer 1 Report

This manuscript shows that pulsatile GnRH stimulation enhances Nppc and Npr2 gene expression in gonadotrope cell lines and identifies transcriptional targets of CNP in the same cells. The study design is very clear and elegant, the study question is original, the topic is physiologically and clinically relevant and the methods are thoroughly described.

Author Response

We thank the reviewer for their extremely positive comments on our work, and note that they have not requested us to make any changes.

Reviewer 2 Report

In this manuscript, Mirczuk et al. studied the role of CNP-signaling in pituitary gonadotrophin cells. Firstly they used multiplex RT-qPCR system to elucidate a comprehensive expression profile of natriuretic peptide-associated genes in mouse endocrine tissues, and then they used two lines of gonadotroph cells and investigated the reciprocal effects of GnRH-axis and CNP-signaling. I think it noteworthy that they showed both the alteration of CNP-signaling components by GnRH stimulation and that of Gn-RH components by CNP-signaling in pituitary gonadotrophin cell line(s). The experiments are well performed and the manuscript is well written. The results and their interpretations seem to be sound.

I only want to make several minor comments and point out a mistake.

Page 5, Figure 1, inset, the representation of RT- reaction is obscure.

Page 6, Figure 2, the value of Npr3, as well as indication in the graph, is unclear. How much?

Page 7, the 2nd line of the legend for Figure 3, A→B and B→C. Further, please add the explanation for Figure 3A in the legend.

Page 9, Figure 4, I think the exhibitions of differences in some graphs are uncertain. As for Nr5a1 gene expression, isn’t there a significant difference between basal and GnRH treatment in the continuous group in LbT2 cells (B) or in aT3-1 cells (C)? As for Nr0b1, isn’t there a difference in the continuous group in LbT2 cells (B) or in the pulsatile group in aT3-1 cells (C)? I think it would be better to mark a significant difference between the GnRH-treated bars of Egrt1 expressions in aT3-1 cells (C).

Author Response

We are extremely grateful for this reviewers complementary and positive critique, and thank them for the few issues they have asked us to address (please see below, our responses are in bold):

Page 5, Figure 1, inset, the representation of RT- reaction is obscure. - We agree with the reviewer, and have removed the inset image in Figure 1 (page 4)

Page 6, Figure 2, the value of Npr3, as well as indication in the graph, is unclear. How much? - We thank the reviewer for pointing this out, and have now adjusted the split in the y-axis in Figure 2 (page 5) to allow these data to be clearly seen.

Page 7, the 2nd line of the legend for Figure 3, A→B and B→C. Further, please add the explanation for Figure 3A in the legend. - Thank you for pointing this out - we have added the description to the figure legend (page 6).

Page 9, Figure 4, I think the exhibitions of differences in some graphs are uncertain. As for Nr5a1 gene expression, isn’t there a significant difference between basal and GnRH treatment in the continuous group in LbT2 cells (B) or in aT3-1 cells (C)? - We have meticulously re-analysed these data, but there is no significant difference in Nr5a1 expression in either cell lines in response to the GnRH stimulation.

As for Nr0b1, isn’t there a difference in the continuous group in LbT2 cells (B) or in the pulsatile group in aT3-1 cells (C)? - Again, we have re-analysed the data, and there is no significant difference in Nr0b1 expression in either cell line.

I think it would be better to mark a significant difference between the GnRH-treated bars of Egrt1 expressions in aT3-1 cells (C). - Thank you for bringing this omission to our attention - we have added the significance annotation between continuous and pulsatile GnRH treatment to this figure (page 8).

Reviewer 3 Report

Mirczuk et al. investigated the expression of natriuretic peptides (NPs) system in pituitary tissue derived from male and female mice as well as in LβT2 and αT3-1 gonadotrope cell lines. The latter were subjected to gonadotropin releasing hormone (GnRH) (continuous vs pulsatile) or C-type natriuretic peptide (CNP) treatment. In pituitary tissues, the authors observed a major expression of Nppc, Npr2 and Furin. They further demonstrated that pulsatile GnRH treatment increases the expression of Nppc and Npr2 in LβT2 and also upregulates cJun, Egr1, Nr5a1 and Nr0b1, known gonadotrope transcription factors.

This work is clear and well organized. The experiments were accurately performed. The strategy to analyze the expression of CNP system under either continuous and pulsatile GnRH stimulation is novel and interesting. These findings also extend previous evidence obtained by the same research group about the role of CNP in the pituitary district.

I suggest to perform few additional experiments which should improve the quality of this manuscript:

Major:

It should be interesting to investigate the level of genes affected by GnRH treatment also at protein level. For example, performing an ELISA assay for CNP (NT-pro CNP) in the conditioned cell media or by western blot analysis for CNP/NP receptors. Please perform the quantification of cGMP in cells after the specific treatments. How the expression of GnRH receptors is affected by GnRH/CNP treatment?

Minor:

-I suggest to analyze the expression of the proprotein convertase subtilisin/kexin type 6 (PCSK6) in order to complete the natriuretic peptide cascade. Since PCSK6 cleaves and activates corin, its expression in pituitary tissue and cells should be weak.

-Figure 3B. Npr3 increases after pulsatile GnRH treatment but also the expression of Npcc is enhanced. It should be expected a decrease expression of Npcc compared to basal, since Npr3 is devoted to the clearance of CNP. Please clarify this aspect.

-Are there gender-differences in gene expression of CNP system in the pituitary tissue?

Author Response

We are very grateful for this reviewers constructive and positive comments with regards to our manuscript, and have addressed their suggestions in the modified manuscript, and in our answers, below, in bold).

It should be interesting to investigate the level of genes affected by GnRH treatment also at protein level. For example, performing an ELISA assay for CNP (NT-pro CNP) in the conditioned cell media or by western blot analysis for CNP/NP receptors. - We agree with the reviewer that it would be interesting to follow up some of these gene expression targets with protein analysis, although this is beyond the scope of our current, comprehensive molecular investigation. There are some issues in relation to measuring CNP protein secreted from these cell lines; firstly, αT3-1 cells possess very few. if any, secretory granules, which significantly reduces the possibility of detecting any secreted peptide released via regulated exocytosis. Secondly, whilst LβT2 have been used to investigate hormone secretion (of LH, in particular), these published studies have required the cells to be primed with steroid (oestrogen) and pulsed with GnRH for several days; were we to adopt this approach, this would represent a completely different treatment paradigm to the one that we have used for our molecular studies. Nevertheless, our future experiments will aim to investigate changes in proteins.

Please perform the quantification of cGMP in cells after the specific treatments. - We have already performed these experiments and include representative data of CNP-stimulated cGMP accumulation in both cells lines, as part of a new Supplemental Figure 3 (page 15). In terms of examining cGMP accumulation in cells treated with continuous or pulsatile GnRH prior to a CNP stimulation, the main issue is that we, and others, have already published that GnRH causes the heterologous desensitisation of GC-B receptors, which would prevent us from measuring CNP-stimulated cGMP accumulation under those conditions.

How the expression of GnRH receptors is affected by GnRH/CNP treatment?  - We have included these data as part of the new Supplemental Figure 3 (page 15); there is no effects on Gnrhr expression in either cell line. We have made reference to these findings in the discussion page 10, line 294).

I suggest to analyze the expression of the proprotein convertase subtilisin/kexin type 6 (PCSK6) in order to complete the natriuretic peptide cascade. Since PCSK6 cleaves and activates corin, its expression in pituitary tissue and cells should be weak. - This is an interesting addition, and one that we would consider in future experiments (particularly our current in vivo experiments); but it is not simple to add primer sets to multiplex RT-qPCR assays, as this requires re-optimisation of the assay conditions.

-Figure 3B. Npr3 increases after pulsatile GnRH treatment but also the expression of Npcc is enhanced. It should be expected a decrease expression of Npcc compared to basal, since Npr3 is devoted to the clearance of CNP. Please clarify this aspect. - Thank you for making this interesting point. We have discussed this on page 10 (line 289-292), in relation to the up regulation of Npr3 likely being a physiological response to the up-regulation in Nppc/Npr2.

-Are there gender-differences in gene expression of CNP system in the pituitary tissue? - Few tissues (adipose, adrenal) show sexual dimorphic differences in Nppc expression, but the pituitary does not (p=0.54, Mann-Whitney).

Round 2

Reviewer 3 Report

The authors addressed all my comments.

Now the manuscript is much improved